# Validation of Existing Clinical Prediction Tools for Primary Aldosteronism Subtyping

**DOI:** 10.3390/diagnostics12112806

**Published:** 2022-11-15

**Authors:** Barbora Kološová, Petr Waldauf, Dan Wichterle, Jan Kvasnička, Tomáš Zelinka, Ondřej Petrák, Zuzana Krátká, Lubomíra Forejtová, Jan Kaván, Jiří Widimský, Robert Holaj

**Affiliations:** 1Centre for Hypertension, 3rd Department of Medicine, General University Hospital and 1st Faculty of Medicine, Charles University in Prague, U Nemocnice 504/1, 128 08 Prague, Czech Republic; 2Department of Anesthesiology, University Hospital Královské Vinohrady and 3rd Faculty of Medicine, Charles University in Prague, Šrobárova 1150/50, 100 00 Prague, Czech Republic; 3Department of Cardiology, Institute for Clinical and Experimental Medicine, Vídeňská 1958/9, 140 21 Prague, Czech Republic; 4Department of Radiology, General University Hospital and 1st Faculty of Medicine, Charles University in Prague, U Nemocnice 504/1, 128 08 Prague, Czech Republic

**Keywords:** adrenal venous sampling, aldosterone-producing adenoma, idiopathic aldosteronism, prediction score, primary aldosteronism, saline infusion test

## Abstract

The new clinical prediction score (SCORE) has been recently proposed for primary aldosteronism (PA) subtyping prior to adrenal vein sampling (AVS). This study aimed to compare that SCORE with previously published scores and their validation using a cohort of patients at our center who had had positive SIT confirming PA and had been diagnosed with either bilateral PA according to AVS or unilateral PA if biochemically cured after an adrenalectomy. Final diagnoses were used to evaluate the diagnostic performance of the proposed clinical prediction tools. Only Kamemura’s model (with a maximum score of 4 points) and Kobayashi’s score (with a maximum score of 12 points) reached 100% reliability for prediction of bilateral PA; however, with sensitivity of only 3%. On the other hand, the values of SCORE = 3 (with sensitivity of 48%), the SPACE score ≥18 (with sensitivity of 35%), the Kobayashi’s score ≤2 (with sensitivity of 28%), and the Kocjan’s score = 3 (with sensitivity of 28%) were able to predict unilateral PA with 100% probability. Furthermore, Umakoshi’s and Young’s models both reached 100% reliability for a unilateral PA with score = 4 and both predictive factors together respectively; however, the sensitivity was lower compared with previous models; 4% and 14%, respectively. None of the clinical prediction tools applied to our cohort predicted unilateral and bilateral subtypes together with the expected high diagnostic performance, and therefore can only be used for precisely defined cases.

## 1. Introduction

The nature of primary aldosteronism (PA) lies in autonomous aldosterone overproduction. Such a condition results in moderate, severe, or often resistant arterial hypertension [1]. Furthermore, elevated aldosterone levels lead to increased potassium losses in urine, thus potentially causing hypokalemia. Such elevation of aldosterone prompts activation of the inflammatory system resulting in collagen depositions in both the myocardial and vascular wall [2], creating myocardial and vascular fibrosis [3,4], and left ventricle and vascular remodeling [5,6,7,8] alongside other pathologies.

PA is known to be the most frequent form of secondary hypertension of the endocrine. The prevalence of PA varies based on the population selection. In a population of non-selected hypertensive patients, it reaches merely about 6% [9,10,11]; however, it rises to even 20% [12,13,14] in patients with moderate or severe hypertension.

Laboratory diagnostics of PA consist of determining the plasma (or serum) aldosterone concentration (PAC) and plasma renin activity (PRA), or direct renin concentration (DRC), and the ratio of PAC to PRA (ARR) or PAC to DRC (ADRR), respectively. Patients with uncertain results should undergo further confirmation tests, such as the saline infusion test (SIT) or captopril challenge test; suppression of aldosterone secretion is determined during this test; if insufficient, PA diagnosis is confirmed [15].

Distinguishing between unilateral and bilateral overproduction of aldosterone is crucial for determining the optimal treatment method as unilateral PA is suitable for surgical treatment by adrenalectomy opposed to the bilateral form, which is to be treated by application of mineralocorticoid receptor blockers [15].

Aldosterone-producing adenoma (APA) is the most common form of the unilateral form of PA, while the bilateral form is usually caused by bilateral adrenal hyperplasia, commonly referred to as idiopathic hyperaldosteronism (IHA) [16]. The reliability of computed tomography (CT) in laterality determination is rather limited due to its relatively low sensitivity for microadenomas and high probability of discovering a non-functioning adenoma [17]. Therefore, adrenal venous sampling (AVS) remains the method of choice when choosing between APA and IHA (including laterality information).

Given how complicated, technically challenging, and expensive [18,19] AVS is, identifying patients with a high probability of either unilateral or bilateral form can be difficult. In such patients, AVS could be omitted, which would be highly beneficial. Nevertheless, the accuracy of the existing clinical prediction scores varies depending on the used databases and, therefore, the clinical applicability of such scores is rather limited [20,21,22,23,24,25,26,27,28,29,30,31,32,33]. In this study, we aimed to compare the accuracy of the latest published prediction score (SCORE) for classifying the PA subtype with other already published prediction methods [34].

## 2. Materials and Methods

### 2.1. Patients

Two cohorts of patients who had been investigated via our tertiary-hospital-based Centre for Hypertension for resistant and severe hypertension, and suspected secondary hypertension were analyzed. In total, 150 patients were enrolled in the development cohort with a suspected diagnosis of PA. The suspicion was based on the minimal absolute concentration of aldosterone of 15 ng/dL and an elevated aldosterone-to-PRA ratio of 30 (ng/dL)/(ng/mL/h). The patients were investigated between November 2003 and June 2011. A total of 138 patients with suspected PA were enrolled in the validation cohort. The suspicion was based on an elevated aldosterone-to-DRC ratio 5.7 (ng/dL)/(ng/L) and on the minimal absolute concentration of aldosterone of 15 ng/dL. These patients were investigated between July 2011 and December 2020. Overnight recumbency and a 2 h standing position were required prior to PRA or DRC and aldosterone levels were measured [35]. Other main forms of secondary hypertension (pheochromocytoma, renal parenchymal disease, renovascular hypertension) or drug-induced hypertension were carefully excluded in all the patients. In order to exclude Cushing syndrome, a 1 mg dexamethasone suppression test was performed in all the patients. Each participant was required to sign written informed consent regarding the study. The original study protocol and protocol amendment were approved by the local Ethics Committee of the General University Hospital in Prague on 26 June 2003 and 21 June 2012, respectively.

### 2.2. Drugs Management

Two weeks prior to admitting the patients to the hospital, their antihypertensive medication influencing the levels of the examined hormones was discontinued (6 weeks, in the case of spironolactone treatment). Oral contraceptives were discontinued for two months in fertile females. Previous antihypertensive treatment was switched to α-blockers (doxazosin) and/or a slow-release non-dihydropyridine calcium channel blockers (verapamil). Potassium substitution was continued in hypokalemic patients [35].

### 2.3. Saline Infusion Test (SIT)

The PAC level was measured before and a rapid administration of 2 liters of saline infusion over 4 h performed. The diagnosis of PA was confirmed when PAC did not decrease below 5 ng/dL.

### 2.4. CT Findings

The CT findings were classified into unilateral lesion, bilaterally normal, and bilateral lesions. Adrenal lesions included nodule (round or oval, with smooth margins, well-defined lesion, measured ≥6 mm in diameter), and hyperplasia (if adrenal gland thickness measured ≥10 mm in diameter) [36,37,38].

### 2.5. Adrenal Venous Sampling (AVS)

Samples from both adrenal veins and the inferior vena cava were analyzed and aldosterone and cortisol concentrations were determined. Adrenocorticotrophic hormone stimulation was not used during the AVS examination. The results were evaluated in accordance with the current expert consensus [18,39,40], according to which a lateralization index >4 together with a selectivity index >2 indicates unilateral aldosterone overproduction. The lateralization index is calculated as [aldosterone(dominant side)/cortisol(dominant side)]/[aldosterone(non-dominant side)/cortisol(non-dominant side)]. The selectivity index is calculated as cortisol(adrenal vein)/cortisol(inferior vena cava) and assessed for both adrenal veins. The samples were evaluated in the patients who were biochemically cured according to the primary aldosteronism surgical outcome (PASO) criteria 6–12 months after unilateral adrenalectomy [41]. AVS, in which the lateralization index was between 2 and 4 was considered as unilateral aldosterone overproduction in certain cases, while other parameters and clinical data were considered, such as the presence of contralateral suppression <1 (which may indicate unilateral PA on the contralateral side), hypokalemia below 3.0 mmol/L, and persistence of uncontrolled hypertension on MRA therapy [40,42].

### 2.6. Laboratory Methods

All endocrine laboratory examinations (PAC, PRA, and plasma cortisol) were performed by radioimmunological analysis, using commercially available kits (Immunotech; Beckman Coulter Company, Prague, Czech Republic) in the dedicated local laboratory prior to June 2011. Since July 2011, all endocrine laboratory examinations (DRC, serum aldosterone, and cortisol) were performed by chemiluminescence assay, using commercially available kits (DiaSorin; Saluggia, Italy) in the central laboratory of the General University Hospital.

### 2.7. Blood Pressure Measurement

An oscillometric device (Omron M6, Shimogyo-ku, Kyoto, Japan) was used for casual blood pressure measurement. The measurement was conducted in a quiet room with the patient’s arm positioned at heart level. Three measurements of blood pressure were performed in a sitting position after 5 min of rest. The resulting value of causal systolic and diastolic blood pressure was calculated as the average of the second and third measurements. The patient’s 24 h blood pressure was measured during hospital stay, using an oscillometric device (SpaceLabs 90207, SpaceLabs Medical, Redmond, WA, USA).

### 2.8. Statistical Analysis

The statistical software R 4.0.3 (R Core Team, Foundation for Statistical Computing, Vienna, Austria) was used for data processing. Parametric data are presented as an average ± standard deviation. Non-parametric data are presented in the format of median and interquartile range. Study groups and subgroups were compared using the t-test for independent samples or the Mann-Whitney U test depending on the distribution of variables. Categorical data were compared using the Pearson chi-square test or Fisher exact test. *p*-values < 0.05 were considered significant. The sensitivity and specificity were calculated as follows: sensitivity = true positive/(true positive + false negative); true positive = the number of unilateral PA patients (diagnosed by AVS and/or surgery) meeting the bypass criteria; false negative = the number of unilateral PA patients (diagnosed by AVS and/or surgery) not meeting the bypass criteria; specificity = true negative/(true negative + false positive), true negative = the number of bilateral PA patients (diagnosed by AVS) not meeting the bypass criteria; false positive = the number of bilateral PA patients (diagnosed by AVS) meeting the bypass criteria.

### 2.9. Comparison with Other Criteria for Bypassing Adrenal Venous Sampling

The key words ‘primary aldosteronism’, ‘adrenal venous sampling’, ‘subtype’ and ‘predict’ were used during a PubMed search to find studies that seemed relevant. Only human studies were considered. PubMed was searched multiple times and the last search was performed on 23 September 2022.

The criteria which had to be met for a model to be included were: confirmation of the subtype diagnosis had to be made based on successful AVS or follow-up after adrenalectomy. The studies that did not fulfill the eligibility criteria were excluded.

## 3. Results

### 3.1. Study Population

One hundred and fifty patients with a definite subtyping diagnosis based on AVS and/or surgery were included in the development cohort and 138 patients in the validation cohort. Unilateral and bilateral PA were diagnosed in 96 and 54 patients in the development cohort and 94 and 44 patients in the validation cohort, respectively. One hundred and seventy-two from the 190 patients with definite subtype diagnosis were diagnosed by PASO and 18 were diagnosed by AVS (no surgery performed). In 150 patients with a unilateral nodule on CT, 126 (84%) were confirmed as unilateral PA by AVS and/or surgery and only one patient had AVS-determined lateralization, which was contralateral to her nodule (<6 mm). The average nodule size on CT was 17.2 ± 5.7 mm in the development cohort and 15.5 ± 7.6 mm in the validation cohort.

The baseline characteristics of the patients are shown in Table 1. Age, BMI, and in-office and 24-hour blood pressure levels were comparable between the unilateral and bilateral subgroups in the development cohort. The subgroup of unilateral PA comprised a significantly higher proportion of females compared with bilateral PA (43% vs. 22%, *p* = 0.01). The patients with unilateral PA had significantly higher baseline PAC (*p* = 0.002), baseline ARR, PAC after the SIT (both *p* < 0.0001), and PAC after the PST (*p* = 0.02). Such patients had also significantly lower serum potassium levels (*p* < 0.001) and, unsurprisingly, significantly higher prevalence of adrenal nodules detected by CT (*p* < 0.0001).

There were no significant differences in the proportion of females, age, BMI, or in-office and 24-h blood pressure levels between patients with unilateral and bilateral PA in the validation cohort. The patients with unilateral PA had significantly higher baseline serum aldosterone (*p* < 0.0001), baseline ARR (*p* = 0.04), serum aldosterone after the SIT (*p* < 0.0001), and serum aldosterone after the PST (*p* < 0.01). They had significantly lower serum potassium levels previously documented and after the SIT (both *p* < 0.0001) and, expectedly, significantly higher prevalence of adrenal nodules detected by CT (*p* < 0.01).

### 3.2. Saline Infusion Test (Development Cohort)

The PAC and PRA values measured before and after the saline infusion are shown in Table 2. The PAC values before infusion were significantly higher than after infusion in all subgroups (*p* < 0.01; *p* < 0.001, respectively). These values were significantly higher in patients with unilateral form than in those with bilateral form (*p* < 0.01 *p* < 0.01). Moreover, in patients with bilateral form, the PRA values before infusion were significantly higher than in patients with APA and patients with bilateral form after infusion (*p* < 0.05).

### 3.3. Validation of the Previously Published Prediction Scores and Models

Finally, 13 studies were selected to be included from a total of 345 articles, which were originally retrieved from PubMed based on their titles. Prevalence of PA subtypes determined using the SCORE based on development or validation datasets was compared to validation dataset and to existing prediction scores. Apart from the work by Song et al., all other authors used PRA to diagnose PA (as we did when testing patients in the development cohort). Only Song et al. used DRC to diagnose PA, as was the case when testing patients in the validation cohort. Therefore, we only tested the prediction model of Song et al. for the data from the validation cohort, while the other prediction models for the data from the development cohort. The three scores (Kobayashi et al., Kocjan et al., and Kamemura et al.) were originally formulated to predict the bilateral subtypes of primary aldosteronism and were therefore reformulated to predict the unilateral form (Table 3).

Using the latest published SCORE [34], the highest score value = 3 had a 100% probability of unilateral PA with sensitivity of 48% in the development cohort. With a score of 0, the probability for bilateral PA was 67% and sensitivity 61%. Fifty five of 150 patients (37%) were in the indeterminate range (Figure 1a).

Using Kobayashi’s recommended score of at least 8 to predict bilateral PA [24], only 25 of 46 (54%) patients in our cohort were correctly identified (sensitivity 46%). Using a score of 3 or less, the probability for unilateral PA was 88% and increased to 97% with a score of 2 or less, with respective sensitivities of 47 and 35%. Fifty three of 150 patients (35%) were in the indeterminate range when a score less than 3 was used, and 69 of 150 patients (46%) were in the indeterminate range when a score less than 2 was used (Figure 1b).

Using Puar’s recommended aldosterone-to-potassium ratio (APR) [25], the probability of unilateral PA was 69% with an APR greater than 100 and increased to 80% with an APR greater than 150, with respective sensitivities of 68 and 49%. With a low APR less than 50, the probability of bilateral PA was 57%, and sensitivity was only 7%. Forty nine of 150 patients (33%) were in the indeterminate range when an APR greater than 100 was used and 84 of 150 patients (56%) were in the indeterminate range when an APR greater than 150 was used (Figure 2a).

Using the SPACE model [27], a score of 18 and more had a 97% probability for unilateral PA and increased to 100.0% with a score of more than 19, and more with respective sensitivities of 40% and 29%. With a score of 2 and less, the probability for bilateral PA was 50% and sensitivity only 6%. One hundred and five of 150 patients (70%) were in the indeterminate range when a score of 18 and more were used and 96 of 150 patients (64%) were in the indeterminate range when a score of 19 and more was used (Figure 2b).

Using Kupers’s model [20], a score of 5 and more had an 89% probability for unilateral PA and sensitivity of 51%, whereas a score of 1 or less had a probability for bilateral PA of 79% and sensitivity of only 28%. Seventy six of 150 patients (51%) were in the indeterminate range (Figure 3a).

Using Nanba’s model [21], a score of 7 and more had an 84% probability for unilateral PA and sensitivity of 40%, whereas a score of 3 or less had a probability for bilateral PA of 57% and sensitivity of 57%. Fifty one of 150 patients (34%) were in the indeterminate range (Figure 3b).

Using Kocjan’s model [22], a score of 2 and more had an 86% probability for unilateral PA and increased to 100.0% with the highest score value = 3 with respective sensitivities of 66% and 100%. With a score of 0, the probability for bilateral PA was 69% and sensitivity 57%. Forty one of 150 patients (27%) were in the indeterminate range when a score of more than 2 was used and 87 of 150 patients (58%) were in the indeterminate range when score = 3 was used (Figure 4a).

Using Kamemura’s model [23], a score of 3 and more had a 96% probability for bilateral PA and sensitivity of 26%, whereas a score of 0 had a probability for unilateral PA of 96% and sensitivity of 20%. Eighty of 150 patients (53%) were in the indeterminate range (Figure 4b).

Using Lee’s model [28], the highest score value = 3 had a 93% probability for unilateral PA and sensitivity of 38%, whereas a score of 0 had a probability for bilateral PA of 94% and sensitivity of 11%. Ninety eight of 150 patients (65%) were in the indeterminate range (Figure 5a).

Using Young’s model [29], both predictive factors together had a 100% probability for unilateral PA and sensitivity of 14%, whereas negativity for both factors had aprobability for bilateral PA of 74% and sensitivity of 83%. Sixty seven of 150 patients (45%) were in the indeterminate range.

Using Umakoshi’s model [31], the highest score value = 4 had a 100% probability for unilateral PA but sensitivity of only 4%, whereas a score 1 or less had a probability for bilateral PA of 86% and sensitivity of 61%. Fifty of 150 patients (33%) were in the indeterminate range (Figure 5b).

Using Kaneko’s model [30], the highest score value = 1 had a 57% probability for unilateral PA and sensitivity of 69%, whereas a score of 0 had a probability for bilateral PA of 69% and sensitivity of 57%.

Using Rossi’s model [32], the highest score value = 3 had a 98% probability for unilateral PA and sensitivity of 15%, whereas a score of 0 had a probability for bilateral PA of 92% and sensitivity of 35%. One hundred and eight of 150 patients (72%) were in the indeterminate range (Figure 6a).

Using the latest published SCORE [34], the highest score value = 3 had a 100% probability of unilateral PA with a sensitivity of 36% in the validation cohort. With a score of 0, the probability for bilateral PA was 55% and sensitivity 55%. Seventy of 138 patients (51%) were in the indeterminate range (Figure 6b).

Using the CONPASS model [33], all predictive factors together had an 89% probability for unilateral PA and sensitivity of 35%, whereas with no predictive factor, theprobability for bilateral PA was 94% and sensitivity 20% in the validation cohort. Ninety seven of 138 patients (70%) were in the indeterminate range.

## 4. Discussion

Based on our retrospective study, we found that none of the prediction scores formulated thus far can be reliably used to rule out bilateral aldosterone overproduction. Kobayashi’s score, which reached 100% reliability with a maximum score of 12 points, seems promising. However, this score value was achieved in only 5 patients from our cohort (that means a sensitivity of only 3%). Conversely, the authors’ recommended value of a score of 8 or more represented only a 54% positive predictive value of bilateral aldosterone overproduction with sensitivity of 46% [24]. Furthermore, Umakoshi’s and Young’s models both reached 100% reliability for unilateral PA with a score = 4 and both predictive factors together, respectively, however the sensitivity was lower compared with that of previous models; 4% and 14%, respectively [29,31]. Also, the reliability of the remaining previously validated prediction models (aldosterone-potassium ratio and SPACE score) for the probability of bilateral aldosteronism was low; specifically, the SPACE score ≤2 reached positive predictive value for bilateral aldosteronism of 50% with sensitivity of 5% and the aldosterone-potassium ratio ≤50 was only 43% with a sensitivity of 3% [27]. Previously not validated predictive scores reached positive predictive values of bilateral overproduction of around 57–79% depending on the selected cut-off values. The sensitivity of these tests for prediction of bilateral aldosteronism did not exceed 60%.

Conversely, some prediction scores showed 100% reliability in predicting unilateral aldosteronism. Apart from our published SCORE = 3 value (with sensitivity of 48%), the similarly constructed Kocjan’s model (with sensitivity of only 28%) also showed 100% reliability [22]. Kobayashi’s score appears to have 100% reliability of the prediction of unilateral overproduction given the patients reach a value of 0–2 (with sensitivity of 28%) [24] and the SPACE score exhibits the same probability in values of 18–20 (with sensitivity of 35%) [27]. The only patient reaching these score values and having been falsely classified as bilateral aldosteronism was a severely hypokalemic woman with unilateral adrenal adenoma who was reclassified as bilateral aldosteronism after adrenalectomy for persistent laboratory findings of primary aldosteronism. Even though the laboratory findings persisted, they diminished. Retrospectively, we speculate, she should have been excluded from the cohort, because theoretically, it could have been a rare combination of unilateral and bilateral overproduction. The same woman was one of the two patients classified as having bilateral aldosteronism because of her score of 7 in the Kupers’s model [20]. The second patient was a man with a unilateral tumor, severe hypokalemia, and renal insufficiency. However, he had an unquestionable bilateral overproduction. The other prediction scores (including previously validated aldosterone-potassium ratio) did not reach any significant predictive power for prediction of unilateral aldosteronism.

Validation of three novel clinical prediction tools for PA subtyping has recently been published by Kocjan et al. [43]. The tools include aldosterone-to-lowest potassium ratio [25], SPACE score [27], and aldosterone concentration, and the aldosterone concentration relative reduction rate after SIT [26]. The originally reported high clinical performance was not reached when the validated tools were applied to their cohort. A major limitation of the published validation was the absence of withdrawal of all the antihypertensive medication potentially affecting the renin-angiotensin-aldosterone system. Continued concomitant beta-blockers therapy can lower aldosterone concentration levels and thus theoretically decrease the lateralization index in patients with lateralized disease. Therefore, some patients with adrenal adenoma and higher aldosterone concentration after SIT may have been misclassified as bilateral overproduction.

The study published by Kobayashi et al. [24] was the only study that respected the generally reported unilateral-to-bilateral PA ratio (specifically, 378 patients with unilateral PA and 912 patients with bilateral PA). Patients with unilateral PA often present with more severe clinical features, such as resistant hypertension or severe hypokalemia, and are thus more likely to be referred to specialized secondary hypertension centers. Be that as it may, a lower percentage of patients with bilateral PA might lead to a selection bias. A similarly disproportionate ratio of bilateral and unilateral PA patients can be seen in our own study.

Different size criteria have been used throughout the studies. Such discrepancy may have also affected the results changing the specificity of unilateral PA. The criterion for the adrenal tumor used in our study was ≥6 mm because 6-mm tumor mass is the minimum size of pathological adrenal tissue that can be detected by our institutional multi-slice CT scanner. Higher cut-off values to detect adrenal tumor were used in other studies: Kobayashi et al. [44]: ≥8 mm, Kobayashi et al. [24]: ≥10 mm, Kupers et al. [20]: ≥8 mm, Kamemura et al. [22,23]: ≥10 mm, and Kocjan et al. [22]: “regardless of size”.

The SIT in PA patients was first widely assessed in the study by Weigel et al. [45]. Patients with PAC >10 ng/dL after the SIT had a higher proportion of unilateral PA compared with the rest of the population (66% vs. 42%; *p* < 0.001). The high predictive value of SIT in distinguishing PA subtypes was recently confirmed in a study by Kaneko et al. [30]. Seated SIT had superior accuracy in subtype diagnosis of PA compared with the dexamethasone suppression test.

One of the explanations for the relatively low sensitivity of some scores for the prediction of bilateral PA may be the relatively higher representation of unilateral hyperplasia as the cause of unilateral overproduction, namely 35% in our group of patients. Patients with unilateral hyperplasia may not have as clearly expressed laboratory findings typical of unilateral overproduction, such as severe hypokalemia or significantly increased aldosterone levels before and after the suppression test. The only patient with a false-positive diagnosis of bilateral PA with a SPACE score = 0 had histologically confirmed unilateral hyperplasia. Even 10 years after adrenalectomy, adrenal hyperplasia on the second side did not develop.

We are aware of the limitations of our study.

Firstly, the APA group showed a higher proportion of females than the IHA group in the development cohort. As the aldosterone levels may be higher in fertile females than in males, maybe due to the high prevalence of the mutation in the KCNJ5 gene in young females with APA [46,47], a higher proportion of females could contribute to the higher aldosterone level found in the APA group.

Secondly, the scoring system is likely specific to our center as there is no universal protocol to diagnose PA across centers with variation in assay methods, cut-off values of screening tests, confirmatory testing, and lateralization criteria for AVS. A prospective multicenter study is therefore required to confirm and validate the accuracy of the presented SCORE.

Thirdly, a lower percentage of patients with bilateral PA, as shown in our study, might lead to a selection bias.

In conclusion, only Kamemura’s model (with a maximum score of 4 points) and Kobayashi’s score (with a maximum score of 12 points) reached 100% reliability for prediction of bilateral aldosteronism; however, with sensitivity of only 3%. On the other hand, the values of SCORE = 3 (with the highest sensitivity of 48%), SPACE score ≥18 (with sensitivity of 35%), Kobayashi’s score ≤2 (with sensitivity of 28%), and Kocjan’s score = 3 (with sensitivity of 28%) were able to predict unilateral aldosteronism with 100% probability. Umakoshi’s and Young’s models both reached 100% reliability for unilateral PA with a score = 4 and both predictive factors together, respectively; however, the sensitivity was lower compared with previous models; 4% and 14%, respectively. The application of the clinical prediction tools to our cohort did not predict bilateral or unilateral subtypes together with the expected high diagnostic performance, although they can still be used for precisely defined cases.

## Figures and Tables

**Figure 1 diagnostics-12-02806-f001:**
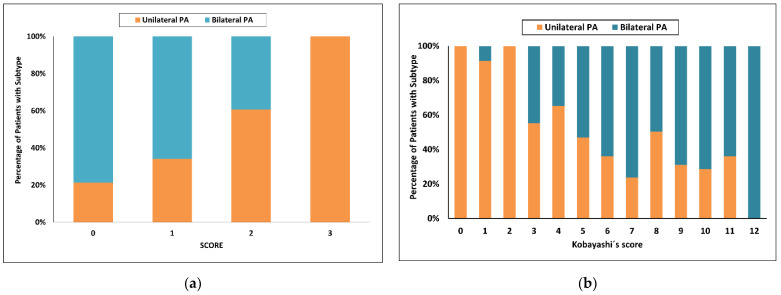
Categorized histograms showing the proportion of primary aldosteronism subtypes depending on the individual scores or models: (**a**) SCORE; (**b**) Kobayashi’s score. PA, primary aldosteronism.

**Figure 2 diagnostics-12-02806-f002:**
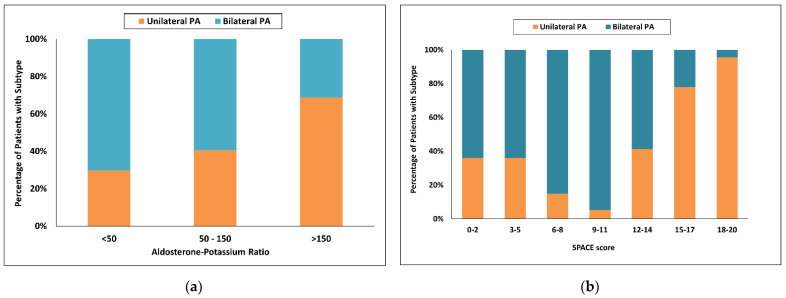
Categorized histograms showing the proportion of primary aldosteronism subtypes depending on the individual scores or models: (**a**) Aldosterone-Potassium ratio; (**b**) SPACE score. PA, primary aldosteronism.

**Figure 3 diagnostics-12-02806-f003:**
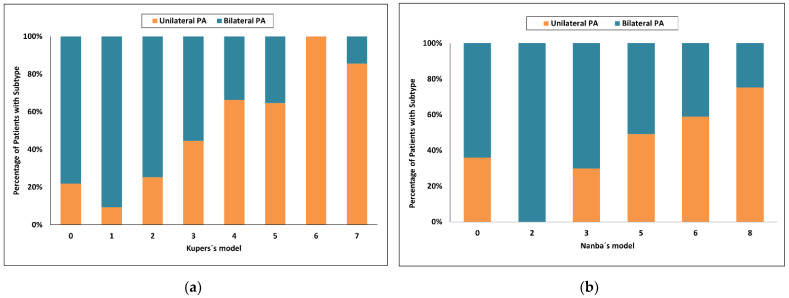
Categorized histograms showing the proportion of primary aldosteronism subtypes depending on the individual scores or models: (**a**) Kupers’s model; (**b**) Nanba’s model. PA, primary aldosteronism.

**Figure 4 diagnostics-12-02806-f004:**
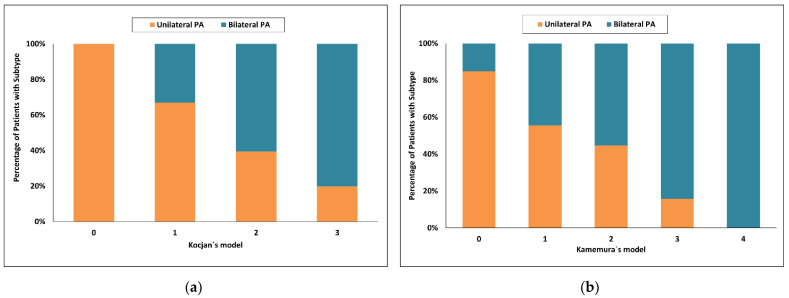
Categorized histograms showing the proportion of primary aldosteronism subtypes depending on the individual scores or models: (**a**) Kocjan’s model; (**b**) Kamemura’s model. PA, primary aldosteronism.

**Figure 5 diagnostics-12-02806-f005:**
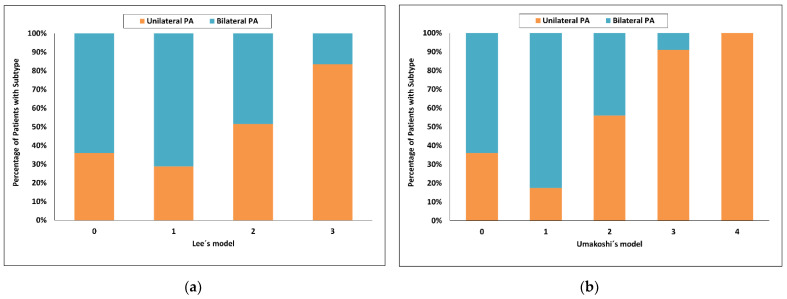
Categorized histograms showing the proportion of primary aldosteronism subtypes depending on the individual scores or models: Lee’s model (**a**), Umakoshi’s model (**b**). PA, primary aldosteronism.

**Figure 6 diagnostics-12-02806-f006:**
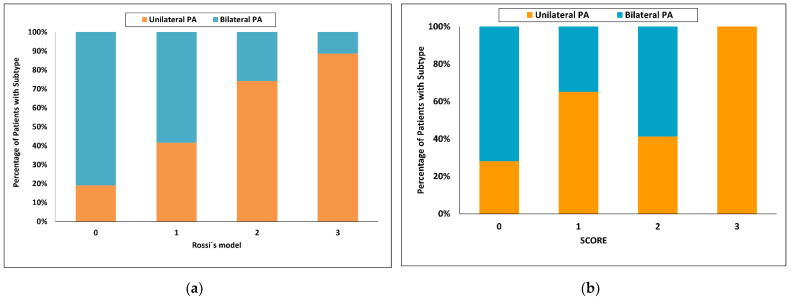
Categorized histograms showing the proportion of primary aldosteronism subtypes depending on the individual scores or models: (**a**) Rossi’s model, (**b**) SCORE validation cohort. PA, primary aldosteronism.

**Table 1 diagnostics-12-02806-t001:** Baseline characteristic of the cohorts.

	Development	Cohort		Validation	Cohort	
	Unilateral PA (*n* = 96)	Bilateral PA (*n* = 54)	*p*-Value	Unilateral PA (*n* = 94)	Bilateral PA (*n* = 44)	*p*-Value
Age (years)	51 (44–58)	52 (47–58)	0.52	49 (41–57)	47 (41–56)	0.55
Females	41 (43%)	12 (22%)	0.01	23 (24%)	13 (30%)	0.53
Body mass index (kg/m^2^)	29 (25–32)	30 (28–32)	0.16	30 (27–33)	31 (29–34)	0.16
Systolic BP (mm Hg)	170 (150–180)	160 (150–170)	0.17	159 (150–170)	155 (145–162)	0.12
Diastolic BP (mm Hg)	100 (90–110)	97 (88–105)	0.12	99 (90–105)	98 (90–103)	0.68
24h systolic BP (mm Hg)	150 (137–161)	150 (137–163)	0.87	150 (140–158)	147 (139–157)	0.48
24h diastolic BP (mm Hg)	93 (85–98)	91 (84–97)	0.69	91 (85–96)	91 (85–98)	0.96
Duration of disease (years)	9 (4–15)	11 (6–16)	0.41	8 (5–12)	7 (3–16)	0.78
Antihypertensive drugs (n)	4 (2–5)	4 (2–6)	0.14	4 (2–4)	3 (2–5)	0.67
Lowest potassium (mmol/L)	3.0 (2.8–3.3)	3.2 (2.9–3.5)	0.11	3.2 (2.9–3.5)	3.6 (3.3–3.9)	<0.0001
Serum potassium (mmol/L)	3.4 (3.1–3.7)	3.7 (3.4–4.0)	0.0009	3.4 (3.2–3.7)	3.9 (3.7–4.1)	<0.0001
eGFR (mL/min/1.73 m^2^)	126 (104–153)	119 (100–133)	0.11	128 (114–165)	152 (110–186)	0.24
Baseline PAC (ng/dL)	29.1 (18.3–53.7)	22.2 (14.1–32.6)	0.002	26.5 (18.7–36.5)	16.8 (13.1–20.8)	<0.0001
Baseline PRA (ng/mL/h)	0.32 (0.20–0.43)	0.35 (0.25–0.53)	0.046	NA	NA	
ARR [ng/dL/(ng/mL/h)]	106 (51–192)	56 (39–91)	<0.0001	NA	NA	
Baseline DRC (ng/L)	NA	NA		1.50 (0.49–2.70)	1.51 (0.50–3.30)	0.62
ADRR [ng/dL/(ng/L)]	NA	NA		13 (6–35)	9 (6–15)	0.04
Adrenal nodules on CT	68 (71%)	8 (15%)	<0.0001	58 (62%)	16 (36%)	0.005
PAC after SIT (ng/dL)	18.0 (12.1–34.8)	11.5 (8.1–16.3)	<0.0001	17.5 (11.7–23.2)	11.0 (7.5–13.9)	<0.0001

Values are shown as medians (interquartile range), or absolute values (percentages). ADRR, aldosterone-to-direct renin ratio; ARR, aldosterone-to-renin ratio; BP, blood pressure; eGFR, estimated glomerular filtration rate = 194 × (SCr)^−1.094^ × (Age)^−0.287^, × 0.739 if female; CT, computed tomography; DRC, direct renin concentration; NA, not applicable; PA, primary aldosteronism; PAC, plasma aldosterone concentration; PRA, plasma renin activity; SIT, saline infusion test.

**Table 2 diagnostics-12-02806-t002:** Results of the saline infusion (development cohort).

	Unilateral PA			Bilateral PA
	All Patients N = 96	Increase in PAC <30% *n* = 31 (32%)	Increase in PAC >30% *n* = 65 (68%)	*n* = 54
Plasma aldosterone concentration (ng/dL)				
Before infusion	35.5 (22.3–56.1) **^,†††^	57.1 (36.1–79.1) ^†††^	30.3 (19.3–44.3) ^††^	28.5 (22.9–38.9) ^†††^
After infusion	18.0 (12.1–34.7) **	27.5 (18.5–52.2)	16.5 (10.2–25.1)	11.5 (8.2–16.3)
%Change from baseline	−42 [−57–(−22)] **	−42 [−51–(−25)]	−41 [−60–(−22)]	−62 [−70–(−44)]
Plasma renin activity (ng/L)				
Before infusion	0.33 (0.24–0.48) *	0.30 (0.23–0.49)	0.34 (0.25–0.47)	0.38 (0.32–0.59) ^†^
After infusion	0.24 (0.17–0.36)	0.22 (0.17–0.40)	0.27 (0.17–0.35)	0.31 (0.21–0.44)
%Change from baseline	−25 [−38–(−13)]	−27 [−37–(−11)]	−22 [−38–(−14)]	−21 [−41–(−10)]

Values are shown as medians (interquartile range) or absolute values and relative values in percent. PA primary aldosteronism. * *p* < 0.05, ** *p* < 0.01, versus bilateral PA, ^†^
*p* < 0.05, ^††^
*p* < 0.01, ^†††^
*p* < 0.001 versus after infusion.

**Table 3 diagnostics-12-02806-t003:** Sensitivity and specificity of published models for prediction of unilateral PA in our development or validation cohort.

References	Model	Sample of Development Cohort	Sample of Validation Cohort	Sensitivity in Our Development Cohort	Specificity in Our Development Cohort
Holaj et al. SCORE [34]	Unilateral nodule ≥ 6 mm, PAC post-SIT > 16.5 ng/dL	UPA = 96 BPA = 54	UPA = 94 BPA = 44	48%	100%
Kupers et al. [20]	Unilateral nodule ≥ 8 mm, serum K^+^ < 3.5 mmol/L and eGFR ≥ 100	UPA = 49 BPA = 38	None	51%	89%
Nanba et al. [21]	PAC ≥ 16.5 ng/dL, ARR post-CCT ≥ 82, and K^+^ ≤ 3.4 mmol/L	UPA = 32 BPA = 39	None	40%	87%
Kocjan [22]	K^+^ < 3.5 mmol/L, PAC post-SIT > 18 ng/dL and unilateral nodule “regardless of size”	UPA = 28 BPA = 39	None	28%	100%
Kamemura [23]	K^+^ < 3.5 mmol/L, unilateral nodule ≥ 8 mm, baseline ARR ≥ 55 and male sex	UPA = 24 BPA = 204	None	20%	96%
Kobayashi (JPAS) [24]	K^+^ < 3.5 mmol/L, baseline PAC ≥ 21.0 ng/dL, unilateral nodule ≥ 8 mm, baseline ARR ≥ 62 and male sex	UPA = 378 BPA = 912	UPA = 202 BPA = 444	35% * 35%	98% * 100%
Puar et al. [25]	PAC to lowest potassium ratio > 15	UPA = 70 BPA = 33	UPA = 48 BPA = 44	50%	78%
Burrello et al. (SPACE) [27]	Unilateral nodule ≥ 8 mm, lowest potassium ≤ 3.9 mmol/L, PAC post-CCT or SIT > 8.9 ng/dL and PAC at screening > 30.3 ng/dL	UPA = 93 BPA = 57	UPA = 40 BPA = 25	40% * 40%	98% * 100%
Lee et al. [28]	Serum K^+^ < 3.5 mmol/L, PAC > 30 ng/dL and unilateral lesion > 7 mm	UPA = 372 BPA = 39	None	38%	93%
Young et al. [29]	Age < 40 and unilateral nodule > 10 mm	UPA = 102 BPA = 84	None	13%	100%
Kaneko et al. [30]	PAC post-SIT > 13.1 ng/dL	UPA = 16 BPA = 48	None	68%	57%
Umakoshi et al. [31]	PAC > 15.9 ng/dL, serum, K^+^ < 3.5 mmol/L, unilateral nodule > 10 mm and age < 35 years	UPA = 258 BPA = 96	None	4%	100%
Rossi et al. [32]	Age < 45 years, K^+^ < 3.6 mmol/L, unilateral nodule ≥ 5 mm	UPA = 131 BPA = 100	None	14%	98%
**References**	**Model**	**Sample of Development Cohort**	**Sample of Validation Cohort**	**Sensitivity in Our Validation Cohort**	**Specificity in Our Validation Cohort**
Holaj et al. SCORE [34]	Unilateral nodule ≥ 6 mm, PAC post-SIT > 16.5 ng/dL	UPA = 96 BPA = 54	UPA = 94 BPA = 44	** 36%	** 100%
Song et al. (CONPASS) [33]	PAC > 20.0 ng/dL, K^+^ ≤ 3.5 mmol/L, PRC ≤ 5 μIU/mL, unilateral nodule ≥ 10 mm	UPA = 268 BPA = 88	UPA = 84 BPA = 117	** 35%	** 89%

ARR, aldosterone-to-renin ratio; BPA, bilateral adrenal hyperplasia; CCT, captopril challenge test; eGFR, estimated glomerular filtration rate; NA, not applicable; PAC, plasma aldosterone concentration; PRC, plasma renin concentration; SIT, saline infusion test; UPA, unilateral primary aldosteronism. * after exclusion of one falsely classified patient; ** sensitivity and specificity in our validation cohort.

## Data Availability

Not applicable.

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
