# Peer review of "Validation of Existing Clinical Prediction Tools for Primary Aldosteronism Subtyping"

_diagnostics, 2022, doi:10.3390/diagnostics12112806_

Round 1
Reviewer 1 Report
The authors of the manuscript compared the accuracy of the prediction score (SCORE) with other previously published prediction methods used for evaluation of different subtypes of primary hyperaldosteronism (PA). The state-of-the-art procedure allowing differentiation between the unilateral and bilateral form of the disease is adrenal vein sampling with subsequent calculation of the aldosterone/cortisol lateralization ratio. However, adrenal vein sampling can be technically challenging, costly, and might not be available in some medical centers. Therefore, the topic chosen by the authors is very important, as it allows omitting the invasive procedure in at least some patients suspected with PA.
The manuscript presents a prospective study on a group of 263 patients with confirmed PA. The authors compared eight non-invasive protocols used in differential diagnostics of unilateral and bilateral PA. Based on their analysis, the authors concluded that none of the evaluated models allows a full exclusion of the bilateral or unilateral form of the disease.
The merits of the study were well planned and executed. My minor remark is:
The authors used the ARR as a criterion for confirming PA; was the minimal absolute concentration of aldosterone of 15 ng/dl considered as well?
However, I was surprised by the similarities between some parts of proposed manuscript and the authors’ previous article entitled “Adrenal Venous Sampling Could Be Omitted before Surgery in Patients with Conn's Adenoma Confirmed by Computed Tomography and Higher Normal Aldosterone Concentration after Saline Infusion Test”, published in Diagnostics. 2022 Jul 15;12(7):1718. doi: 10.3390/diagnostics12071718
In both works, sections such as the introduction, materials and methods, or the table characterizing the studied population are almost identical. It is understandable that both papers can be focused on the same group of patients, but these sections of the manuscript should be rephrased and rewritten. At the same time, all previously published materials (like the table or figure) should cite the original article.
Major revisions are required.
Author Response
Thank you for your valuable comments
Comments and Suggestions for Authors
The authors of the manuscript compared the accuracy of the prediction score (SCORE) with other previously published prediction methods used for evaluation of different subtypes of primary hyperaldosteronism (PA). The state-of-the-art procedure allowing differentiation between the unilateral and bilateral form of the disease is adrenal vein sampling with subsequent calculation of the aldosterone/cortisol lateralization ratio. However, adrenal vein sampling can be technically challenging, costly, and might not be available in some medical centers. Therefore, the topic chosen by the authors is very important, as it allows omitting the invasive procedure in at least some patients suspected with PA.
The manuscript presents a prospective study on a group of 263 patients with confirmed PA. The authors compared eight non-invasive protocols used in differential diagnostics of unilateral and bilateral PA. Based on their analysis, the authors concluded that none of the evaluated models allows a full exclusion of the bilateral or unilateral form of the disease.
The merits of the study were well planned and executed.
My minor remark is:
The authors used the ARR as a criterion for confirming PA; was the minimal absolute concentration of aldosterone of 15 ng/dl considered as well?
- This criterion for confirming the diagnosis of PA was added in the new version of manuscript.
However, I was surprised by the similarities between some parts of proposed manuscript and the authors’ previous article entitled “Adrenal Venous Sampling Could Be Omitted before Surgery in Patients with Conn's Adenoma Confirmed by Computed Tomography and Higher Normal Aldosterone Concentration after Saline Infusion Test”, published in Diagnostics. 2022 Jul 15;12(7):1718. doi: 10.3390/diagnostics12071718
In both works, sections such as the introduction, materials and methods, or the table characterizing the studied population are almost identical. It is understandable that both papers can be focused on the same group of patients, but these sections of the manuscript should be rephrased and rewritten. At the same time, all previously published materials (like the table or figure) should cite the original article.
Major revisions are required.
- We tried to paraphrase all similarities in the new version of manuscript.
Reviewer 2 Report
In this study, the authors aimed to compare the accuracy of the latest published prediction score (SCORE) for classifying the PA subtype with others already published prediction methods. Based on their retrospective study, they found that none of the prediction scores formulated thus far can be reliably used to predict bilateral or unilateral aldosterone overproduction. The study is interesting and clinically important. However, I have some concern about the following points:
Major points:
When comparing the models, the probability and sensitivity of unilateral PA and bilateral PA are given, but the specificity is not mentioned. When predicting unilateral and bilateral, more consideration should be given to specificity to avoid unnecessary surgery for bilateral PA patients and missing the chance of surgery for unilateral PA patients.
In table 4, the details (criteria) of each model should be provided. Using sensitivity and specificity to show the results in table 4 would make it easier for the reader to understand.
Results shown in Figure 1 and table 4 were partially overlapped.
The author should provide in the methods part what “key terms” did they use to search for the published models.
This paper does not cover all published models for predicting PA subtype. For example: 1) Hikaru Hashimura et al. Saline suppression test parameters may predict bilateral subtypes of primary aldosteronism. Clin Endocrinol (Oxf). 2018 Sep;89(3):308-313. 2) Hironobu Umakoshi et al. Accuracy of adrenal computed tomography in predicting the unilateral subtype in young patients with hypokalaemia and elevation of aldosterone in primary aldosteronism. Clin Endocrinol (Oxf). 2018;88(5):645-51. 3) Seung Hun Lee et al. Diagnostic Accuracy of Computed Tomography in Predicting Primary Aldosteronism Subtype According to Age. Endocrinol Metab (Seoul). 2021;36(2):401-12. 4) Gian Paolo Rossi et al. Feasibility of Imaging-Guided Adrenalectomy in Young Patients With Primary Aldosteronism. Hypertension. 2022;79(1):187-95. 5)Ying Song et al. Development and validation of model for sparing adrenal venous sampling in diagnosing unilateral primary aldosteronism. J Hypertens. 2022 Sep 1;40(9):1692-1701.
the sample size of patients with bilateral PA is small and the author should acknowledge this as a limitation of the study.
The response to posture was not much relevant to the aim of the study, and it should be deleted.
Since July 2011, all endocrine laboratory examinations were performed by chemiluminescence assay, DRC results should be provided.
Minor points
Line 49: “Insufficient suppression of aldosterone secretion during the saline infusion test (SIT) is taken as a final confirmation of the PA diagnosis”. SIT is only one of the confirmatory tests recommended by the guidelines, and the other tests such as captopril challenging test should also be mentioned when introduce the diagnosis of PA.
Line 79: “Each patient underwent a 1 mg dexamethasone suppression test to rule out the Cushing syndrome”. Are there any PA patients were complicated with Cushing syndrome? If so, how to interpret the AVS results in those patients?
Line80: “Each participant signed a written informed consent with the study”. This was a retrospective study, how could each participant sign a written informed consent with the study?
Author Response
Thank you for your valuable comments
Comments and Suggestions for Authors
In this study, the authors aimed to compare the accuracy of the latest published prediction score (SCORE) for classifying the PA subtype with others already published prediction methods. Based on their retrospective study, they found that none of the prediction scores formulated thus far can be reliably used to predict bilateral or unilateral aldosterone overproduction. The study is interesting and clinically important. However, I have some concern about the following points:
Major points:
When comparing the models, the probability and sensitivity of unilateral PA and bilateral PA are given, but the specificity is not mentioned. When predicting unilateral and bilateral, more consideration should be given to specificity to avoid unnecessary surgery for bilateral PA patients and missing the chance of surgery for unilateral PA patients.
Sensitivity and specificity are the two statistical measures most commonly used to assess the performance of an alternative test against the gold standard. Sensitivity and specificity measure how well a test classifies subjects who truly have/do not have the outcome of interest, respectively. Positive and negative predictive value reflect the proportion of positive and negative test results, respectively, that are truly positive and truly negative. In contrast to sensitivity and specificity, which are generally considered inherently stable for a given diagnostic test, positive predictive value and negative predictive value are highly dependent on pre-test probability, wherein positive predictive values increase with increased disease prevalence, and negative predictive values increase with decreased disease prevalence. Therefore, positive predictive value was replaced by sensitivity in the new version of the manuscript.
In table 4, the details (criteria) of each model should be provided. Using sensitivity and specificity to show the results in table 4 would make it easier for the reader to understand.
In the new version of the manuscript, the sensitivity was added to table 4, or new 3.
Results shown in Figure 1 and table 4 were partially overlapped.
After the content changes in table 4, respectively new 3 no longer overlaps information.
The author should provide in the methods part what “key terms” did they use to search for the published models.
Keywords to search for studies dealing with prediction models have been added to the method section.
This paper does not cover all published models for predicting PA subtype. For example: 1) Hikaru Hashimura et al. Saline suppression test parameters may predict bilateral subtypes of primary aldosteronism. Clin Endocrinol (Oxf). 2018 Sep;89(3):308-313. 2) Hironobu Umakoshi et al. Accuracy of adrenal computed tomography in predicting the unilateral subtype in young patients with hypokalaemia and elevation of aldosterone in primary aldosteronism. Clin Endocrinol (Oxf). 2018;88(5):645-51. 3) Seung Hun Lee et al. Diagnostic Accuracy of Computed Tomography in Predicting Primary Aldosteronism Subtype According to Age. Endocrinol Metab (Seoul). 2021;36(2):401-12. 4) Gian Paolo Rossi et al. Feasibility of Imaging-Guided Adrenalectomy in Young Patients With Primary Aldosteronism. Hypertension. 2022;79(1):187-95. 5)Ying Song et al. Development and validation of model for sparing adrenal venous sampling in diagnosing unilateral primary aldosteronism. J Hypertens. 2022 Sep 1;40(9):1692-1701.
Thank you for reminding of the papers that were not mentioned in the original version of the manuscript. All five listed papers were also included in the analysis in the new version of the manuscript including the recently published work by Song et al. (see Table 3).
The sample size of patients with bilateral PA is small and the author should acknowledge this as a limitation of the study.
The note regarding the proportionally lower number of patients with BPA has been removed from discussion to study limitations of the study.
The response to posture was not much relevant to the aim of the study, and it should be deleted.
The paragraph devoted to the postural test (including the table) was deleted in the final version of the manuscript.
Since July 2011, all endocrine laboratory examinations were performed by chemiluminescence assay, DRC results should be provided.
We reviewed all the works listed in Table 3. With the exception of the work by Song et al. all other authors used plasma renin activity to diagnose PA (as we did when testing patients in the development cohort). Only Song et al. used DRC to diagnose PA as we did when testing patients in the validation cohort. Therefore, we only tested the prediction model of Song et al. on data from the validation cohort, while the other prediction models on data from the development cohort. This fact is stated in the results paragraph in the new version of the manuscript.
Minor points
Line 49: “Insufficient suppression of aldosterone secretion during the saline infusion test (SIT) is taken as a final confirmation of the PA diagnosis”. SIT is only one of the confirmatory tests recommended by the guidelines, and the other tests such as captopril challenging test should also be mentioned when introduce the diagnosis of PA.
This fact was added to the introduction of the new version of the manuscript.
Line 79: “Each patient underwent a 1 mg dexamethasone suppression test to rule out the Cushing syndrome”. Are there any PA patients were complicated with Cushing syndrome? If so, how to interpret the AVS results in those patients?
There was no patient with primary aldosteronism complicated concurrently by Cushing's syndrome in our group.
Line80: “Each participant signed a written informed consent with the study”. This was a retrospective study, how could each participant sign a written informed consent with the study?
Part of the informed consent signed by all patients was also consent to the later use of the collected data for statistical purposes.
Round 2
Reviewer 1 Report
The authors not only rephrased and rewrote the initial version of the manuscript, but also modified the research data they presented. New patients were added to the studied group with number of participants now reaching 288. New subgroups of patients suspected with PA based on ARR ≥30 (n=150) and aldosterone/DRC ratio (n=138) were distinguished.
The authors assessed additional prediction models previously used in evaluation of suspected PA. Based on their analysis, none of the models allowed full exclusion of bilateral PA.
My minor remarks for this stage of peer review are:
1. Table 1 requires some corrections. PAC and baseline PRA values were moved throughout the lines making the table difficult to read.
It seems that the second use of “ARR” might be in fact referring to “ADRR” (plasma aldosterone concentration/direct renin concentration).
2. In line 102, the “DRC” abbreviation appears for the first time, therefore it should be explained. In this fragment of the manuscript, the authors should clearly indicate the ADRR cutoff considered typical for PA (the sentence in lines 100-103 should be rewritten to clearly state the chosen criterion). More so, used units should be unified (ng/dL, ng/mL?).
Author Response
To Reviewer #1:
Thank you for your valuable comments
The authors not only rephrased and rewrote the initial version of the manuscript, but also modified the research data they presented. New patients were added to the studied group with number of participants now reaching 288. New subgroups of patients suspected with PA based on ARR ≥30 (n=150) and aldosterone/DRC ratio (n=138) were distinguished.
The authors assessed additional prediction models previously used in evaluation of suspected PA. Based on their analysis, none of the models allowed full exclusion of bilateral PA.
My minor remarks for this stage of peer review are:
- Table 1 requires some corrections. PAC and baseline PRA values were moved throughout the lines making the table difficult to read.
- The table has been fixed, in order to make it more organized.
It seems that the second use of “ARR” might be in fact referring to “ADRR” (plasma aldosterone concentration/direct renin concentration).
- The aldosterone/direct renin concentration ratio is now referred to as ADRR in the new version of the manuscript.
- In line 102, the “DRC” abbreviation appears for the first time, therefore it should be explained. In this fragment of the manuscript, the authors should clearly indicate the ADRR cut-off considered typical for PA (the sentence in lines 100-103 should be rewritten to clearly state the chosen criterion).
- The abbreviation DRC is now explained on the line 51 in the new version of the manuscript.
More so, used units should be unified (ng/dL, ng/mL?).
- The units used throughout the manuscript have been unified.

Reviewer 2 Report
1) Line 58: ARR is not “the golden standard of laboratory diagnostics” for PA.
2) Line 102: Should 5.7 (ng/dL)/(ng/mL) be 5.7 (ng/dL)/(ng/L)?
3) Should the unit for DRC be ng/L?
4) Were all the unilateral PA were diagnosed by PASO or some were diagnosed by AVS? The criteria of unilateral PA and bilateral PA should be added in the method.
5) As CT is an important criterion in most models, the methods and criteria for CT lesions should be detailed (e.g is unilateral nodule means no nodule/hyperplasia on the contralateral side? What is the definition of nodule? What is the definition of hyperplasia?).
6) Table 1. ”Adrenal nodule on CT” is ambiguous. What is the averaged diameter of the nodule? Bilateral or unilateral? How about the hyperplasia? The detailed CT results should be given.
7) Table 3. the results in the validation model should be provided separately instead of *; and models which could be tested should be done both in development and validation cohorts.
8) Table 3. for model from Rossi et al, the age should be <45 years, not ≥ 45 years
9) Table3. Dose unilateral nodule in each model means no nodule/hyperplasia on the contralateral side?
10) It is hard to understand the figure1. What does the abscissa of Figure 1 mean? If it is the score of each model, some models do not use scores (e.g Young’s model, CONPASS model)
Author Response
To Reviewer #2:
Thank you for your valuable comments
1) Line 58: ARR is not “the golden standard of laboratory diagnostics” for PA.
- The formulation has been fixed on the lines 50-52 in the new version of the manuscript.
2) Line 102: Should 5.7 (ng/dL)/(ng/mL) be 5.7 (ng/dL)/(ng/L)?
- The unit has been changed in the new version of the manuscript.
3) Should the unit for DRC be ng/L?
- The unit has been used in the new version of the manuscript.
4) Were all the unilateral PA were diagnosed by PASO or some were diagnosed by AVS? The criteria of unilateral PA and bilateral PA should be added in the method.
- The diagnostic criteria for unilateral and bilateral form are listed in the 2.5 paragraph.
- Eighteen patients with bilateral PA were diagnosed based on AVS and the remaining 172 were diagnosed based on PASO.
5) As CT is an important criterion in most models, the methods and criteria for CT lesions should be detailed (e.g is unilateral nodule means no nodule/hyperplasia on the contralateral side? What is the definition of nodule? What is the definition of hyperplasia?).
- CT criteria for nodules and hyperplasia are listed in the paragraph 2.4.
6) Table 1. ”Adrenal nodule on CT” is ambiguous. What is the averaged diameter of the nodule? Bilateral or unilateral? How about the hyperplasia? The detailed CT results should be given.
- The average nodule size on CT was 17.2 ± 5.7 mm in the development cohort and 15.5 ± 7.6 mm in the validation cohort.
7) Table 3. the results in the validation model should be provided separately instead of *; and models which could be tested should be done both in development and validation cohorts.
- This requirement is rather difficult to be met. The models are far from optimal to be tested using the validation cohort. Some of the earlier models contain the values of plasma renin activity (PRA) (Kobayashi et al., Nanba et al., Kamemura et al.) or the ARR value which contains PRA as predictive factors. Conversely the CONPASS model contains the direct renin value, which is why it cannot be tested using the development cohort, as the cohort uses PRA as renin value.
8) Table 3. for model from Rossi et al, the age should be <45 years, not ≥ 45 years
- This mistake has been fixed in the new version of the manuscript.
9) Table3. Dose unilateral nodule in each model means no nodule/hyperplasia on the contralateral side?
- Yes.
10) It is hard to understand the figure1. What does the abscissa of Figure 1 mean? If it is the score of each model, some models do not use scores (e.g Young’s model, CONPASS model)
- Figure 1 has been divided into 6 separate double pictures for it to be more easily understandable. The pictures contain Young’s model and the CONPASS model has been omitted in the new version of the manuscript.
